# Peripheral Glycolysis in Neurodegenerative Diseases

**DOI:** 10.3390/ijms21238924

**Published:** 2020-11-24

**Authors:** Simon M. Bell, Toby Burgess, James Lee, Daniel J. Blackburn, Scott P. Allen, Heather Mortiboys

**Affiliations:** Sheffield Institute for Translational Neurosciences, University of Sheffield, Sheffield S10 2HQ, UK; t.burgess@sheffield.ac.uk (T.B.); james.lee@sheffield.ac.uk (J.L.); d.blackburn@sheffield.ac.uk (D.J.B.); s.p.allen@sheffield.ac.uk (S.P.A.); h.mortiboys@shef.ac.uk (H.M.)

**Keywords:** glycolysis, Alzheimer’s disease, Parkinson’s disease, motor neuron disease, fibroblasts, red blood cells, muscle

## Abstract

Neurodegenerative diseases are a group of nervous system conditions characterised pathologically by the abnormal deposition of protein throughout the brain and spinal cord. One common pathophysiological change seen in all neurodegenerative disease is a change to the metabolic function of nervous system and peripheral cells. Glycolysis is the conversion of glucose to pyruvate or lactate which results in the generation of ATP and has been shown to be abnormal in peripheral cells in Alzheimer’s disease, Parkinson’s disease, and Amyotrophic Lateral Sclerosis. Changes to the glycolytic pathway are seen early in neurodegenerative disease and highlight how in multiple neurodegenerative conditions pathology is not always confined to the nervous system. In this paper, we review the abnormalities described in glycolysis in the three most common neurodegenerative diseases. We show that in all three diseases glycolytic changes are seen in fibroblasts, and red blood cells, and that liver, kidney, muscle and white blood cells have abnormal glycolysis in certain diseases. We highlight there is potential for peripheral glycolysis to be developed into multiple types of disease biomarker, but large-scale bio sampling and deciphering how glycolysis is inherently altered in neurodegenerative disease in multiple patients’ needs to be accomplished first to meet this aim.

## 1. Introduction

Neurodegenerative diseases (ND) are a group of nervous system conditions characterised pathologically by the abnormal deposition of protein throughout the brain and spinal cord. The neurodegenerative conditions represent a wide variety of clinical presentations, the most common being Alzheimer’s disease (AD), Parkinson’s disease (PD) and Amyotrophic Lateral Sclerosis (ALS). AD and dementia alone is thought to effect approximately 50 million people worldwide, with PD, the second most common ND, effecting approximately 6.1 million people worldwide [1]. Neurodegeneration is not just seen in the classical ND as many other conditions that affect the nervous system such as multiple sclerosis and epilepsy display neurodegenerative change as the disease progresses [2]. Although presentations of the ND can be quite different, they share common pathophysiological traits. Unfortunately, no disease modifying therapies for the most common ND have yet been identified.

One common pathophysiological change seen in all ND is change to the metabolic function of nervous system and peripheral cells. These alterations involve changes to the metabolic function of mitochondria, alterations in lipid metabolism, and changes to the metabolism of glucose via the glycolytic pathway [3]. Glycolysis is the conversion of glucose to pyruvate or lactate which results in the generation of adenosine triphosphate (ATP) and metabolites for metabolic pathways including the citric acid cycle and fatty acid metabolism [4]. Glycolysis is common to all cell types within the human body and allows for a more rapid generation of ATP than oxidative phosphorylation (OxPHOS), the main metabolic pathway by which mitochondria produce ATP. Changes to glycolysis in peripheral non-nervous system cells have been reported in many ND including AD, PD and MND [5,6,7,8]. This is an interesting finding as not only does it highlight the fact that many ND have a systemic effect, but also highlights how a key cellular metabolic pathway is altered across multiple ND. Understanding how glycolysis in peripheral cells changes in ND is important as it may allow for the development of biomarkers of ND from easily accessible cells. Developing new therapeutics for different ND that target peripheral glycolysis may also lead to the development of treatments that could modify the disease course.

In this review article, we briefly discuss the normal function of the glycolysis pathway, and then describe changes seen to glycolysis in the main ND focusing mainly on changes seen in peripheral non-nervous system cells. This review article will end with the challenges and benefits of developing biomarkers directed at peripheral cell glycolysis in ND. 

## 2. Glycolysis

The glycolysis pathway is a combination of several enzymes and co-enzymes that metabolise glucose into pyruvate or lactate. The conversion of glucose to pyruvate generates ATP via the actions of the phosphoglycerate kinase (PGK) and pyruvate kinase (PK) and consumes ATP via the action of hexokinase (HK) and phosphofructokinase (PFK). Nicotinamide adenine dinucleotide (NADH) is created in the glycolytic pathway via the action of glyceraldehyde-3-phosphate dehydrogenase (GA3PDH) and consumed by lactate dehydrogenase [9]. Overall glycolysis produces a net of 4 molecules of ATP and two molecules of NADH. There are three main rate-limiting enzymes in the glycolytic pathway, they are HK, PFK and PK, with PFK being the main rate limiting enzyme [9,10,11]. Glucose-6-phosphate the product of the action of HK on glucose can be utilized by other metabolic pathways including the pentose phosphate shunt (PPS) and gluconeogenesis.

PPS is a branch of glycolysis where glucose-6-phosphate can be converted to ribose 5-phosphate, and via this conversion produces two molecules of nicotinamide adenine dinucleotide phosphate (NADPH). Ribose 5-phosphate is a sugar that can be used in the synthesis of nucleic acids, whereas NADPH is an important reducing equivalent that helps to protect the cell from oxidative stress [12]. It is estimated that 10–20% of glucose consumed by a cell is metabolized via the PPS. Gluconeogenesis and glycogenolysis are the metabolic pathways by which glycogen, a cellular glucose storage molecule, can be created and converted back into glucose when glucose delivery to a cell is reduced. 

Glycolysis is the metabolic pathway by which red blood cells (RBC), and astrocytes mainly produce ATP, and is a more immediate source of ATP than oxidative phosphorylation (OxPHOS). Glycolysis is utilized to great effect by the brain. At times of increased brain activity glycolysis can offer a rapid supply of ATP, even though glycolysis produces fewer molecules of ATP per glucose molecule consumed when compared to OxPHOS [13]. Glycolysis can be directly observed in the human brain using radioactive forms of glucose such as 2-[^18^F] fluoro-2-deoxy-D-glucose (FDG). Using 2-[^18^F]fluoro-2-deoxy-D-glucose positron emission tomography (FDG-PET), a reduction in glucose metabolism has been seen in both ageing and AD [14], although this shift accounts for glucose consumed in both glycolysis and other metabolic pathways. Figure 1 shows the glycolytic metabolic pathway and the fates of the different products of glucose metabolism. The following sections will discuss how glycolysis and its associated metabolic pathways are affected in ND.

## 3. Glycolytic Changes in Alzheimer’s Disease

Alzheimer’s disease (AD) is both the most common neurodegenerative disease and most prevalent form of dementia worldwide [1]. It is calculated that around 50 million people globally have dementia [15,16], with between 60–80% of cases thought to be due to AD [17]. In 2016 dementia was the fifth leading cause of death worldwide [1], and in 2018 it was estimated that the total cost of dementia care to the world economy was $1 Trillion dollars [16]. Pathologically AD is characterised by the build-up of extracellular plaques that are mainly composed of the amyloid precursor protein (APP) product amyloid beta (Aβ), and intracellular aggregates composed mainly of the cytoskeletal protein tau [18]. 

In AD there is a reduction in aerobic glycolysis in brain areas susceptible to amyloid deposition [19], and in brain areas where high levels of tau accumulation is seen [20]. The change in brain glucose metabolism seen on FDG-PET imaging in AD is relatively specific, and reveals a temporal-parietal pattern of low glucose metabolism, which can be used to distinguish AD from other neurodegenerative diseases such as dementia with Lewy bodies and fronto-temporal dementia [21]. Post mortem (PM) brain samples have identified PFK activity to be elevated in the temporal and frontal lobes of patients with sporadic AD [22], and potentially these changes in PFK function occur in glial cells, as PFK enzyme colocalizes with glial fibrillary acidic protein (GFAP) [23]. LDH and PK enzymes have both been shown to be significantly increased in the temporal and frontal lobes [23] of PM AD brain specimens. One study investigating neurons from PM samples using mass spectroscopy quantified glycolytic enzyme protein expression in AD showing a down regulation of PK, enolase, aldolase A, aldolase C, and up regulation of glyceraldehyde-3-phosphate dehydrogenase (G3PDH) [24]. Interestingly, most PM studies show a general upregulation in glycolytic enzyme protein, but brain imaging studies suggest a low glucose uptake by the brain in AD, which is potentially an area of contradiction between the two research techniques. These differences might be explained by the reduction of glucose transporters (GLUT1 and 3) also reported within the brain as the disease progresses [25,26]. Increased enzyme expression may be a response to reduced glucose availability caused by a reduction in the GLUT1 & 3 receptors. Increased glycolytic enzyme protein is also thought to reflect deficits in mitochondrial function seen in AD as the disease progresses [27,28]. 

There are extensive changes seen to the metabolism of glucose in cell types that are not characteristically thought to express disease changes in AD. Fibroblasts from patients with AD have been shown to have both increased [29] and decreased [30] glucose utilization when compared to control fibroblast lines. Both these papers suggest that altered glucose utilization is explained by abnormal mitochondrial function leading to a reduction in the OxPHOS. An initial increase in glucose metabolism in AD fibroblasts is seen because glycolytic ATP production initially compensates for OxPHOS deficits, but eventually overall glucose utilisation is reduced because of mitochondrial inefficiency leading to decreased OxPHOS [29,30]. In a more recent paper in which glucose uptake, glycolytic rate and enzyme expression has been measured in AD fibroblasts, an overall increase in the rate of glycolysis is seen [8]. The same paper also investigates mitochondrial function and again suggests that decreased mitochondrial OxPHOS capacity is the reason for increased glycolysis [8]. Interestingly this paper shows that in sporadic AD fibroblasts the increase in the rate of glycolysis, is not matched with an increase in glucose uptake by fibroblasts. This may explain the apparent differences observed in glucose utilisation observed in previous papers [29,30]. If a fibroblast cannot increase glucose uptake to match cellular glycolysis demands, then glucose utilization in AD fibroblasts may not be uniform across different experiments. This finding of reduced glucose uptake also mirrors the reduction of glucose uptake seen within the AD brain [21,26]. Interestingly, higher blood glucose levels have been associated with higher brain glucose and reduced glycolytic flux and reduced GLUT3 receptors on neurons as AD progresses [26]. This highlights some of the common mechanisms of glycolytic dysfunction that occur both in the nervous system and periphery in AD. 

When the enzymes of glycolysis have been measured in AD fibroblasts increases in the gene expression of the allosteric activator of PFK; 6-phosphofructo-2-kinase/fructose-2,6-biphosphatase 3 (PFKFB3), and LDH have been shown [8], but a reduction of the activity of HK has also been reported [31]. The increased glycolytic enzyme activities have been reported in sporadic AD fibroblasts whereas decreased HK activity was reported in familial AD fibroblasts, but only from members of one of two families examined [31]. Potentially the increased expression of PFKFB3 maybe a compensatory mechanism for a reduction in HK activity as this enzyme can regulate the rate of glycolysis through its actions on PFK. Although in the paper which showed a reduction in HK activity no increase in PFK activity was reported [31,32]. Alternatively, the reduced HK activity may represent increased glycolysis as the enzyme is inhibited by its product fructose-6-phosphate. This observation also highlights that variability in enzyme function may be mutation dependant in familial AD. 

Glycolytic capacity and extracellular lactate have been studied in sporadic AD fibroblasts. A significant decrease [6] and significant increase in glycolytic capacity has been reported in AD fibroblasts [8]. Where a reduction in glycolytic capacity is seen fibroblasts were assessed initially in a glucose free media, whereas an increase in glycolytic capacity was seen in fibroblasts grown in a glucose rich media. This suggests that glucose storage may be affected in AD fibroblasts, or again may be related to the rate at which AD fibroblasts can uptake glucose. As already described AD fibroblasts appear to not be able to increase the rate at which they uptake glucose [8]. Therefore, when AD fibroblasts are grown in a glucose free media intracellular glucose supplies may be depleted quicker than in non-AD fibroblasts leading to a reduction in glycolysis rate. Lactate increase [8,30] and decrease [6] has been reported in AD fibroblasts, but this difference may be explained by whether the lactate measured is intra or extracellular. Increased intracellular and decreased extracellular lactate could be explained by the observed increase in glycolysis that appears to be compensating for decreased mitochondrial function. The lactate may be utilized by the cell and therefore is not released into the media. 

The study of RBCs in AD has also shown alterations to the rate of glycolysis. Increases in the activities of several enzymes of the glycolysis pathway including HK, PFK, bisphosphoglycerate mutase and bisphosphoglycerate phosphatase have been reported in RBCs from patients with AD [33]. This change in glycolysis function is associated with a reduction in 2,3-diphosphoglycerate (2,3-DPG) levels, an enzyme integral to the ability of RBCs to release oxygen into surrounding tissues [33]. It has been postulated that this observation may to some extent explain the reduced oxygen consumption observed in the brains of patients with AD [10]. These changes seen in the RBCs of patients with AD may be a consequence of increased Aβ exposure. Both increased RBC aging and increased activation of glycolysis enzymes is reported when RBCs are exposed to high levels of Aβ [34,35]. It has been shown that Aβ increases the activity of the RBC Na/K-ATPase pump in animal models [35], and that AD patient RBCs have increased Na/K-ATPase pump activity [33]. This has led to the interpretation that increased glycolytic enzyme activity in RBC in AD results from an increased demand for ATP generated through glycolysis to maintain the Na/K-ATPase pump [36,37]. This leads to the interesting question of how RBCs would be exposed to increased levels of Aβ. Both platelets [38,39] and fibroblasts [40,41] have been shown to have increased Aβ secretion in AD which may lead to the reported increases in serum Aβ seen in patients at risk of developing AD [42]. The RBCs exposure to Aβ may also occur when the RBCs circulate through the central nervous system (CNS), as the blood vessels of the brain are known to accumulate amyloid [43], and there is evidence of increased passage brain amyloid in to the blood in AD. Alternatively, these abnormalities in RBCs glycolysis may be inherent and exacerbated by the increased amyloid load seen in AD, or be the consequence of mitochondrial DNA mutations. The understanding of abnormal glycolysis in RBCs would benefit from future studies correlating brain amyloid load with glycolytic dysfunction. This would help to determine if RBCs have changes to glycolysis that are independent of Aβ accumulation. It is interesting that both RBC and brain tissue have been shown to have an up regulation of glycolytic enzyme activity, specifically the PFK enzyme in AD but fibroblasts do not appear to have an alteration in glycolytic enzyme activity. This is potentially an effect of sample size as all studies investigating peripheral glycolysis in AD are relatively small. The exposure to amyloid and the effect this has on both the rate of glycolysis and mitochondrial function may also explain why this glycolytic pathway appears to be more altered in RBCs and brain tissue, with these cell types having greater glycolytic demands than fibroblasts. Table 1 illustrates the changes to glycolysis rate and enzymatic function seen in peripheral cells in AD.

Abnormalities of enzymes in the PPS have also been reported in both the RBC of patients with AD with reductions in glutathione peroxidase and glutathione S-transferases [44]. Both antioxidant enzymes require glutathione generated via the PPS enzyme glutathione reductase activity and NADPH synthesis [45]. This exposes this cell type to reactive oxygen species (ROS) damage which again may impair the ability to deliver oxygen to the tissues of the body. In B-lymphocytes taken from people with AD increased uptake of 2-deoxyglucose, an analogue of glucose that cannot be metabolised past the HK step of glycolysis, has been identified suggesting altered glycolysis [46]. White blood cell glycolysis in AD has not been investigated to the same level as RBCs so it remains unknown to what extent this pathway is altered in this cell type. Reductions in HK1 activity have been reported in leukocytes from patients with familial AD [31] but little else is known about glycolytic enzymatic change in AD white blood cells.

AD also appears to have indirect effects on the glycolytic pathway in different non-nervous system tissues. The synthetic function of the liver is altered in AD patients with changes to the ratio of aspartate transaminase (AST) to alanine transaminase (ALT) correlating with impairments in cognitive functioning and CSF Aβ levels [47]. Reductions in ALT effect the concentration of pyruvate in certain tissues as this enzyme catalyses the creation of pyruvate from α-ketoglutarate and L-alanine. AST has a role in the metabolism of glutamate, which can also be used as a substrate for glycolysis therefore potentially effecting glycolytic rates in cells. Work on an APP mouse model of AD has shown that the liver and kidneys of this animal have reduced levels of glucose, lactate, glucose-6-phosphate and fructose-6-phosphate, which also suggests that these cell types have altered glycolysis in AD [48]. Blood plasma taken from people with AD has been shown to have reduced levels of amino acids that are used as carbon sources for the tricarboxcyclic acid cycle (TCA) when glycolysis is impaired [49] suggesting reduced glycolysis in blood plasma. AD patient blood plasma itself has been shown to increase the rate of glycolysis when applied to microglial cells [50]. This change in microglial glycolysis is thought to be caused by activated compliment inhibiting mitochondrial function [50]. Activated compliment inhibiting mitochondrial function may also explain why increased RBC glycolysis has been seen in the studies mentioned above. 

Together these studies highlight that glycolysis is altered in many non-nervous system cell types in AD, both exposure to amyloid any mitochondrial function are likely to explain elements of these alterations. Although the activity of ATP dependent ions pumps in RBC is also thought to affect the rate of glycolysis. What is yet to be identified is if these changes are a result of increased brain amyloid production, increased systemic levels of amyloid, reduced cellular glucose delivery or glycolytic enzymatic dysfunction independent of amyloid build up. 

## 4. Glycolytic Changes in Parkinson’s Disease

Parkinson’s disease (PD) is the second most common ND after AD. It is characterised by α-synuclein aggregation and the loss of dopaminergic neurons in the Substantia Nigra Pars Compacta (SN), resulting in a combination of motor and non-motor symptoms. Clinically the disease is characterised by the classical triad of rigidity, “pill-rolling” tremor and bradykinesia [51]. The prevalence of this chronic disease is growing faster than AD, doubling between 1990 and 2015, and predicted to double again by 2040 [52]. The growing number of cases poses an alarming economic concern with direct medical costs in the US estimated to be in excess of $25 billion in 2017 [53]. The majority of treatments available for PD revolve around the replacement of the lost dopamine as dopaminergic neurons degenerate.

Glycolysis and glucose uptake abnormalities have been described within the brain of people living with PD. A study in patients using PET and single photon emission computed tomography demonstrated widespread decreased cortical glucose consumption in the early stages of PD [54,55]. Glycolytic changes have also been observed in toxin-induced and genetic models of PD. 6-hydroxydopamine (6OHDA) negatively regulated aerobic glycolysis in rat SN by downregulating the expression of glycolytic enzymes, including HK2, PK, PKD1 and LDH, while upregulating the expression of pyruvate dehydrogenase. This reduction in aerobic glycolysis and altered enzyme expression could be reversed by treating the rats with hydrogen sulphide, which also prevented Parkinsonian behaviour and the loss of DA neurons. Protection was mediated by the upregulation of leptin, however, the mechanism of protection by leptin was not discovered [56]. Observation of zebrafish deficient in the familial PD associated protein PTEN-induced kinase 1 (PINK1) uncovered an upregulation of TigarB, the ortholog of human TP53 Induced Glycolysis Regulatory Phosphatase (TIGAR). TigarB is a bisphosphatase that lowers fructose-2,6-bisphosphate levels thus inhibiting glycolysis via a reduction in PFK activity. Knockdown of TigarB rescued dopaminergic neurons [57]. Further research identified TIGAR positive Lewy bodies in the SN of sporadic PD patients [58]. These results all suggest a decrease in glycolytic activity and glucose consumption within the PD brain.

The majority of PD research in peripheral cells focuses on α-synuclein toxicity, mitochondrial dysfunction and oxidative stress. However, explorative studies into fibroblasts, peripheral blood mononuclear cells and myocytes have revealed alterations in glycolytic activity. Patient-derived fibroblasts have become a popular model for Parkinson’s research; they have been utilised in mechanistic and drug screening studies. However, it is proposed that fibroblasts must be cultured in glucose-free media to study mitochondrial dysfunction [59,60,61,62] in PD. Fibroblasts cultured in glucose media have a net ATP production via glycolysis and oxidative phosphorylation. Whereas fibroblasts cultured in media containing galactose alone do not have a net production of ATP from glycolysis, unmasking aberrations in mitochondrial function [59]. This suggests that glycolysis may play a protective role in PD pathology, and like in AD, compensates for abnormal mitochondrial function. Research in PINK1 KO mouse embryonic fibroblasts (MEF) has shown that glycolysis and glucose uptake were reported to be 2-fold higher and the PPS activity is reduced [59]. Elevated glycolysis seen in these PINK KO MEFs, is associated with an increase in HK2, glyceraldehyde-3-phosphatase and PDK1, and the glucose receptors GLUT1, GLUT3, protein and mRNA levels. The abundance of the transcription factor Hypoxia-inducible factor 1 (HIF1) increases due to stabilisation by ROS and thus upregulates the expression of these glycolysis related proteins [63]. Similarly, increases in GLUT1, GLUT3 and HIF1 protein were observed in skeletal muscle of KO mice although this was not quantified. KO mice were also hypoglycaemic and hyperlactatemic suggesting upregulated glycolysis [63,64]. 

Conversely, Dues et al. reported a reduction in glycolytic activity in patient-derived fibroblasts from patients with PD [60], and no change in glycolysis has been reported in PARK2 mutant and sporadic PD fibroblasts [62,65]. Measurements of basal and stimulated glycolysis showed that extracellular acidification rate did not differ between PD and control patient-derived fibroblasts [62,65]. Similarly, a study in myocytes determined that the ratio of oxidative phosphorylation to glycolysis was consistent between PINK1 knockout mice and controls [66]. Therefore, results are mixed for the effect PD has on peripheral glycolysis. These differences reported in glycolysis may reflect that variability of mitochondrial dysfunction in PD and the fact that this will determine the rate of cellular glycolysis. The apparent discrepancies could also be explained by differences in experiment paradigms as cellular glycolytic capacity can be altered dependant on whether cells are maintained in glucose containing of glucose free media. Several reports have shown that cellular glycolytic capacity can be altered by the presence or absence of media glucose [6,8,67]. It must also be considered that the different mutations that cause PD studied in these papers may have differing effects on glycolysis. 

Study of peripheral blood mononuclear cells (PBMCs), lymphocytes and monocytes, has identified a significant increase in glycolysis in PD and also in patients with Rapid Eye Movement-Sleep Behaviour Disorder (RBD), which is a major risk factor for PD [68]. Extracellular acidification rate analysis revealed that PD and RBD PBMCs have higher glycolysis, glycolytic activity and glycolytic reserve than control groups [69]. While no difference in oxygen consumption was observed. Further investigation revealed that two glycolytic enzyme transcript levels, PDK1 and LDH B, were significantly reduced in the PD population, possibly driving an increase in the TCA cycle activity. Interestingly, glucose uptake and GLUT1 expression did not change suggesting the upregulation of glycolysis is independent of glucose availability, again a similarity with AD research on fibroblast cells. Despite excluding possible drivers, the exact cause and mechanism of increased glycolysis in PBMCs remains undiscovered. This paper did not investigate the role of the Na/K-ATPase pump, which is postulated in AD RBCs to effect glycolysis rates [30]. These findings indicate that the upregulation of glycolysis in easily accessible PBMCs could be a potential early biomarker of PD [69]. However, extensive mechanistic studies of glycolytic activity in fibroblasts, myocytes and PBMCs must be carried out to eradicate the conflicting results that are present in the current literature. It has been suggested from predictive models of metabolism change in PD that SN glucose uptake does not increase with progression of PD [70]. This finding would appear to reflect what is seen in PD PBMCs as both glucose uptake and GLUT transporter levels do not change. A lack of increase in glucose uptake appears to be common to both PD and AD, in both the nervous system and periphery. This is an interesting finding as it highlights a metabolic pathway alteration that is common to both diseases. Future research may benefit from comparing glucose uptake in both ND, as differences between the two conditions may determine the varied clinical phenotypes. Table 1 highlights the glycolytic alterations seen in peripheral cells in PD. 

Regardless of the contradictions observed in peripheral cells, neurons and the brain, glycolysis has been suggested as a possible therapeutic target in PD. Several studies have noted that artificially altered gene expression can upregulate glycolysis and rescue dopaminergic neurons. Overexpression of miRNA-7 increased ATP production, glycolysis and lactate production in SH-SY5Y cells and protected against methyl-4-phenylpyridinium (MPP+) toxicity [71]. Similarly, overexpression of the HK2 protects against rotenone-induced cell death in SH-SY5Y cells and cultured mouse brainstem neurons by increasing glycolysis and HK2 association with mitochondria [72]. Degeneration of dopaminergic neurons in the SN of MPP+ and rotenone-induced mouse models was also abolished by overexpression of HK2 and behavioural deficits were diminished [73]. Additionally, pharmacological interventions are being explored. The drug Meclizine could be repurposed to potentially treat PD. It is an antihistamine agent that is neuroprotective against 6OHDA in both SH-SY5Y and rat primary cortical cultures. Meclizine activates the glycolytic enzyme PFK, thus elevates glycolysis [74]. However, terazosin, a licenced drug for treating benign prostatic hyperplasia and hypertension, appears to be the most promising glycolysis-enhancing treatment. It interacts with the glycolytic enzyme PGK1 by a quinazoline motif, activating PGK1 and consequently increasing pyruvate production. In toxin-induced genetic in vivo and in vitro models of PD, the decline in ATP levels and dopaminergic neurons was partially attenuated by terazosin. While motor deficits were reduced and in some delayed treatment experiments motor deficits were improved in all in vivo models. Furthermore, terazosin is a licensed drug, therefore human data could be analysed and compared to tamsulosin, another benign prostatic hyperplasia treatment that does not target PGK1. Analysis of two PD databases showed terazosin, and other drugs containing a quinazoline motif, slowed the progression of motor deficits or reduced the risk of specific motor, nonmotor and complication phenotypes compared to tamsulosin. Drugs containing the quinazoline motif reduced the incidence of PD compared to tamsulosin, however, this did not reach significance [75]. These promising findings resulted in terazosin progressing into three clinical trials, NCT03905811, NCT04551040 and NCT04386317 although no results have been published yet [75,76]. These aim to assess safety, target engagement and efficacy in preventing PD. However, safety is a major concern due to terazosin’s hypotensive effect. Patients with PD are prone to postural hypotension resulting in falls and terazosin may exacerbate this issue causing more frequent falls [76]. 

Research has established glycolysis as a possible target of disease-modifying treatments in PD, but the understanding of glycolysis in peripheral cell types and as a biomarker remains largely unexplored and conflicting. Nevertheless, the identification of enhanced glycolysis in PBMCs alludes to a promising avenue for biomarker discovery. 

## 5. Glycolytic Changes in Amyotrophic Lateral Sclerosis

ALS is characterised by progressive loss of upper motor and lower motor neurons, with a prevalence of 2 per 100,000. The disease is incurable and typically fatal within 2–5 years of symptom onset [77]. Clinically, depending on the site of presentation of symptoms people with ALS develop progressive muscle weakness, which eventually leads to respiratory failure and death. Roughly 90% of ALS cases are characterised as sporadic, with the 10% of familial ALS (fALS) cases most commonly caused by mutations in the C9orf72, SOD1 and TDP-43 genes [78]. Limited treatments are available for people living with ALS. Riluzole an anti-glutamatergic agent has been shown to extend survival by an average of 3 months in patients with ALS [79], but few other treatments are widely available. 

The importance of metabolic dysfunction in ALS has been extensively studied in recent years, with recent evidence suggesting a link between metabolic dysfunction, disease pathogenesis and disease progression rates [80,81,82]. Mitochondrial dysfunction has been described in multiple ALS models in both central nervous system (CNS) and peripheral tissues, with mitochondrial membrane depolarisation, uncoupling and increased ROS production leading to electron transport chain (ETC) dysfunction and impaired ATP generation [7,80,83,84,85,86,87]. This dysfunction of the mitochondrial ETC is accompanied by an increased dependence on glycolysis or lipid metabolism, hypothesised to be in compensation for impaired ATP production [88,89]. These metabolic adaptations offer potential therapeutic targets, in addition to opportunities for biomarker development [90].

Whilst mitochondrial dysfunction has been the primary focus of research into ALS related metabolic changes in the CNS, alterations in glycolysis can also occur, with evidence emerging that the pathway is a potential target for ameliorating ALS bioenergetic dysfunction [88,89]. Human brain FDG-PET studies have shown mixed results, but the majority of studies show reduced glucose metabolism in the front-parietal cortical regions, and premotor cortical areas [91]. Human spinal cord and muscle FDG-PET studies have suggested hypermetabolism in ALS, although again hypermetabolism in this situation refers to glucose consumed in multiple metabolic pathways [92]. Astrocytes from patients with the most common familial ALS mutation in the gene *C9orf72* display a strong preference for glycolysis [93]. Moreover, supplementation with inosine, which can provide carbon for glycolysis via the PPS, has been shown to significantly increase ATP production in *C9orf72* astrocytes. Mutations in the deoxyribonucleic acid (DNA) binding protein TDP-43, which disrupt autophagy by production of cytoplasmic inclusion bodies [94], results in increased levels of phosphoenolpyruvate and pyruvate in *Drosophila* models of ALS and a concomitant decrease in the pentose phosphate metabolite ribulose. These changes were also observed in iPSC-derived motor neurons and spinal cord homogenates from patients with TDP-43 mutations. Intriguingly, locomotor function increased and survival was extended when flies were fed high sugar diets, or expression of GLUT3 was increased in neurons and glia [88]. These findings highlight upregulation of glycolysis as a potential neuroprotective mechanism and therapeutic target in the CNS.

In muscle, glycolytic alterations have been described in numerous disease models. In the SOD1-G86R and SOD1-G93A mouse models of ALS, a shift from glucose to lipid use was observed, leading to enhanced aerobic exercise performance [95]. This shift was caused by increased Pyruvate dehydrogenase lipoamide kinase isozyme 4 (PDK4) expression, leading to reduced PDH activity and a concomitant shift to glycogen synthesis. This may in part have been driven by the acidosis that develops during ALS, which can cause a shift to glycogen synthesis over lactate production [89,95,96]. FDG-PET studies of SOD1-G93A mice suggested that glucose retention was significantly higher in muscle, although activity of PFK, the rate limiting enzyme in glycolysis, was significantly decreased [97]. Increased glucose uptake is hypothesised to be a response to increased hexose-6-phosphate dehydrogenase activity, an enzyme from the PPS, highlighting how increased glucose uptake/retention is not indicative of increased glycolysis. A concomitant increase in reactive oxygen species production was seen, suggesting utilisation of the PPS may be to promote an antioxidant response.

Co-administration of iron chelation with a high energy diet, a therapy that improves motor function and extends survival in SOD1-ALS mouse models, increased expression of glycolytic enzymes, highlighting the potential mechanism for therapeutic action [98]. In Drosophila models, TDP-43 expression solely in muscles produced locomotor deficits in larvae, but high sugar diet and GLUT3 overexpression did not improve these deficits [88], suggesting targeted glycolytic upregulation in the CNS rather than the periphery may provide metabolic therapeutic benefits in ALS.

Research into immune cell metabolism in ALS is limited despite the role of the immune system in ALS pathogenesis [99,100]. Current literature has focused on lymphoblastoid cell lines (LCL’s), with a recent metabolic analysis performed on LCL’s obtained from patients with sporadic ALS, in addition to patients with familial mutations in FUS, SOD1 and TDP-43 [101]. Bioenergetic flux analysis showed that mitochondrial respiration was increased in all LCL ALS cell models but glycolysis (measured by extracellular acidification rate (ECAR)) was only altered in the SOD1 LCL’s. However, the SOD1 lines displayed a higher baseline ECAR, indicating a significant TCA cycle component. These findings suggest modifications to the rate of glycolysis may be dependent on genetic background in immune cells. Given the role of macrophage infiltration and microglial activation in ALS, further research into metabolism in these immune cells in ALS may help identify further therapeutic targets for ameliorating disease progression [99,100].

Fibroblast cultures from ALS patients have revealed alterations in glycolysis, potentially as a compensation for mitochondrial dysfunction. The SOD1 mutation I113T, which produces cytoplasmic inclusion bodies, led to an increase in ATP produced from glycolysis compared to controls [85]. In sporadic ALS (sALS) fibroblasts, dependence on glycolysis for ATP production was also observed: whilst control fibroblasts shifted to mitochondrial respiration as they aged, sALS fibroblasts did not, indicating an altered response to aging [84]. Moreover, sALS fibroblasts display reduced capacity to upregulate glycolysis when exposed to the Complex I inhibitor rotenone [86]. However, an extensive characterisation of fibroblasts from 171 sALS patients found an increase in glycolysis in tandem with mitochondrial respiration, suggesting general upregulation of metabolism [7]. These findings highlight the systemic nature of metabolic dysfunction in ALS, which should be considered when investigating production of biomarker metabolites and effectiveness of new therapies.

The use of metabolic biomarkers in diagnosis and monitoring of ALS is an as yet underutilised opportunity. Systemic metabolic dysfunction may modify levels of metabolites in easily collected body fluids, such as blood and urine, offering the potential to predict prognosis and monitor disease progression in a minimally invasive manner. In SOD1 G93A mice, acidosis develops in the blood, spinal cord and brain stem towards the end stage of the disease [96]. Serum changes in pH were attributed to changes in strong ion gap [97], a measure which takes into account common strong and weak acids in the blood, indicating an unidentified anion is responsible for blood acidosis. 

The extracellular domain of p75 neurotrophin receptor (p75-ECD) is a potential urinary biomarker for ALS [101,102,103,104]. During neuronal stress and injury, the extracellular domain of the receptor can be cleaved [105]. The receptor regulates glucose uptake, with p75-knockout mice displaying increased glucose uptake [106]. The urine concentration of p75-ECD was significantly higher in SOD1 G93A mice relative to WT mice. Moreover, p75-ECD levels were elevated in ALS patient urine relative to both healthy and neurological controls, indicating its specificity as an ALS biomarker [102,103]. P75-ECD concentration also increased during disease progression, with higher baseline levels indicating poorer prognosis [101,104]. Elevated urine p75-ECD levels may be indicative of impaired glucose uptake as the extracellular domain of the receptor is cleaved, highlighting an impaired metabolic state in ALS pathogenesis. Table 1 displays studies that describe peripheral cell glycolytic alterations in ALS.

## 6. Challenges to Developing Peripheral Biomarkers of Glycolysis in ND Disease

A biomarker is defined as a biological parameter which can be objectively measured to characterise a normal or pathological biological process [94]. When studying disease, or in the clinical realm, biomarkers can be used to identify a condition, track a disease response to intervention, or help to prognosticate the eventual outcome of a patient with a particular condition. As deficits in glycolysis seem to be common to several peripheral cell types and present in all ND studied in this review there is potential that a metabolic biomarker could be developed for ND which could be used in any of the three scenarios described above.

The type of peripheral cell used to develop a peripheral glycolytic biomarker of ND may depend on the type of biomarker that one wishes to develop. With our current level of knowledge a RBC based biomarker may be more appropriate to track disease course once a diagnosis has been established as glycolytic changes to RBCs in PD and AD appear to be similar. A muscle based glycolytic biomarker maybe more useful to diagnose ALS as abnormalities in muscle glycolysis have as yet to be reported in AD and PD. The variable nature of fibroblast glycolytic change seen in ND may represent different stages of ND and so may be useful in developing a prognostic biomarker. For all three ND reviewed in this paper biomarkers of disease are already commonplace in clinical practice so newly developed peripheral glycolytic biomarkers could be combined with present ones to further specify disease diagnosis. 

Before a peripheral biomarker of ND is developed the evidence base for peripheral glycolytic changes needs to be developed. Most studies investigating glycolysis in peripheral cells in ND have small numbers of patients. This is a particular problem when studying sporadic forms of ND as they are very likely to have multifactorial causes for developing ND and therefore only a subgroup of patients may have glycolytic changes in peripheral cells. The interlinked relationship that glycolysis has with other cellular metabolic pathways also needs further deciphering. Several of the changes reported in this review are thought to be as a consequence of mitochondrial dysfunction rather than a primary deficit in glycolysis. As glucose utilization can be measured in both the brain and periphery it may be that a glycolysis biomarker could be developed to track the compensatory changes that happen to cellular metabolism as a consequence of mitochondrial dysfunction. Understanding what changes in glycolysis are inherent to the pathway would be important to establish before a biomarker could be developed. Techniques such as clustered regularly interspaced short palindromic repeat (CRISPR) genome editing and the use of mitochondrial cybrids [95] would help to identify glycolytic specific changes in ND. Large cohort studies have already been developed for ND such as A Multicentre Biomarker Resource Strategy in ALS (AMBRoSIA) [96] which has the potential to further develop our understanding of peripheral glycolysis in ND by assessing many different cell types from large groups of patients. Finally, before a glycolysis biomarker can be developed for ND, it is important to identify how changes to glycolysis correlate with disease stage, progression and treatment response.

## 7. Conclusions

In the three major ND covered in this review there is evidence that glycolysis is altered through the disease course. In all three diseases altered glycolysis is seen as a consequence of abnormal mitochondrial function, but may also be related to protein aggregation and alterations to ATP dependent ion pumps. Developing alterations in glycolysis into a biomarker has certain challenges, but as glycolytic changes are present in many cell types that are easy to access such as RBC this should be considered. It may be that glycolysis abnormalities in peripheral cells are first seen as a biomarker of ND as opposed to a specific sub-type of ND. As abnormalities in glycolytic function effect multiple ND then development of a tracking biomarker may be a more appropriate starting point for development. Further work is needed in large cohorts of patients with ND to understand fully the contribution that abnormal peripheral glycolysis has to the overall disease process, and whether this can be used to the advantage of both clinicians and researchers for the development of future treatments for these diseases.

## Figures and Tables

**Figure 1 ijms-21-08924-f001:**
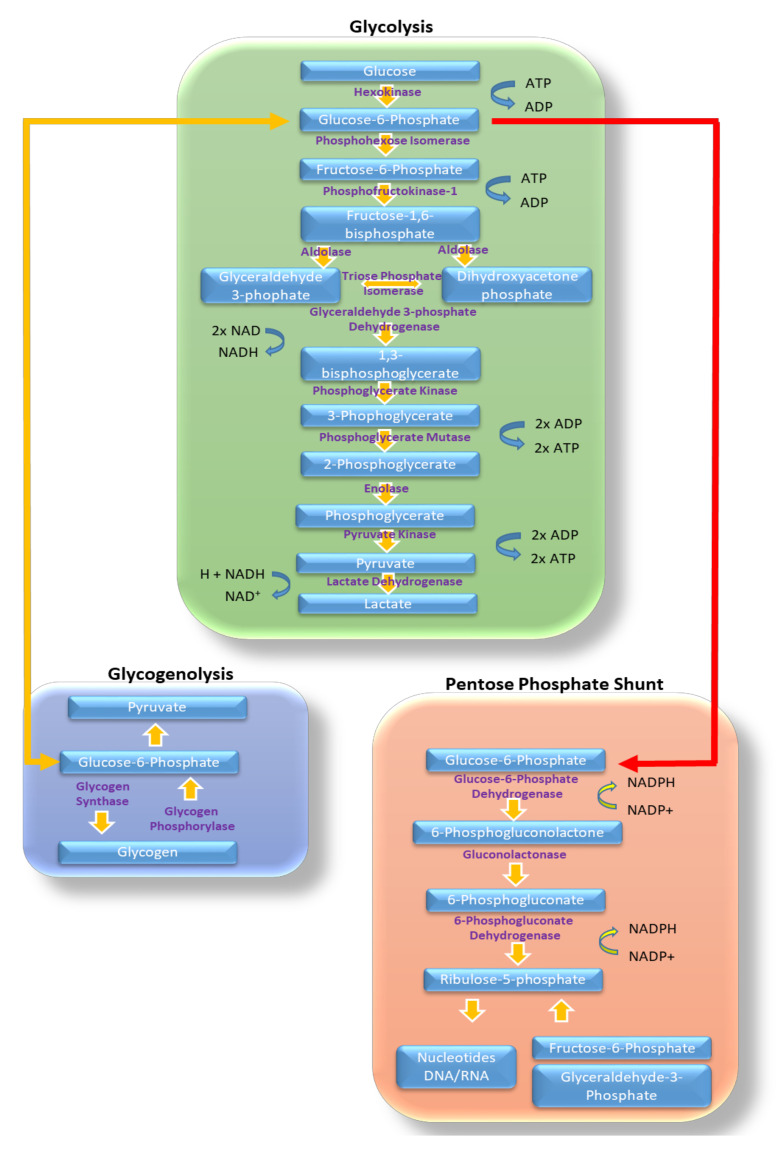
Glycolysis and other glucose metabolic pathways. Highlighted in this figure are the different enzymes and substrates involved in the metabolism of glucose via, glycolysis (green box), glycogenolysis (blue box) and in the pentose phosphate shunt (orange box). Sites of ATP, NADPH, NADH, and ADP generation are shown. Sites of transfer of glucose products between metabolic pathways are highlighted with yellow arrows for movement into glycogenolysis and red for movement though the pentose phosphate shunt.

**Table 1 ijms-21-08924-t001:** Summary of peripheral glycolysis studies in neurodegenerative disease.

Study	Glycolysis Rate	Glycolytic Enzymatic Change	Disease Type	Cell Type	Number of Participants
**Bell et al., 2020** [6]	(−) (↓ Glycolytic Capacity)	Not Assessed	sAD	Fibroblast	10 sAD 10 Controls
**Sonntag et al., 2017** [8]	↑	↑ LDH ↑ PFKFB3 (gene expression)	sAD	Fibroblast	10 AD 13 Young Controls 7 Old Controls
**Sorbi et al., 1995** [29]	↓	Not assessed	sAD fAD	Fibroblast	7 fAD 19 sAD 20 Controls
**Allen et al., 2014** [85]	↑	Not assessed	fALS (SOD1)	Fibroblast	3 Controls 3 SOD1
**Allen et al., 2015** [84]	↑	Not assessed	sALS	Fibroblast	6 sALS 10 Controls
**Raman et al., 2015** [86]	↓ after complex I inhibition	(-) PGK1	sALS	Fibroblast	11 sALS 15 Controls
**Konrad et al., 2017** [7]	↑	Not assessed	sALS	Fibroblast	91 controls 171 sALS
**Sorbi et al., 1990** [31]	Not assessed	↓ HK (in fAD), (-) PFK, (-) LDH	sAD & fAD	Fibroblast Leukocyte	5 Controls, 5 sAD, and 5 fAD (Leukocytes) 8 Controls, 6 fAD (Fibroblasts)
**Sims et al., 1985** [30]	↑	Not assessed	sAD fAD	Fibroblast	6 AD 6 Controls
**Sims et al., 1986** [32]	Not assessed	(-) PFK	sAD	Fibroblast	8 Controls 8 sAD
**Deus et al., 2020** [60]	↓	Not assessed	sPD	Fibroblast	5 Control 5 sPD All male
**Milanese et al., 2019** [59]	Not assessed, theorises ↑	Not assessed	sPD	Fibroblast	21 Controls 47 sPD
**Zanellati et al., 2015** [54]	No change	Not assessed	PARK2 mutant PD	Fibroblast	4 Controls 4 fPD
**Requejo-Aguilar et al., 2014** [63]	↑ glycolysis and glucose uptake	↑ GLUT1, HK-2, PDK1 and glyceraldehyde-3-phosphatase protein and mRNA, ↑ GLUT3 mRNA	PINK1 -/- Mice	Fibroblast (mouse embryonic fibroblasts)	
**Ambrosi et al., 2014** [57]	No change		sPD	Fibroblast	7 Controls 11 sPD
**Kaminsky et al., 2013** [33]	↑	↑ HK, ↑ PFK, ↑ bisphosphoglycerate mutase ↑ bisphosphoglycerate phosphatase	sAD Non-AD dementia	RBC	12 AD 13 non-AD 14 Aged matched controls 14 Young Controls
**Carelli-Alinovi et al., 2019** [34]	↑	Not assessed	Controls exposed to Aβ	RBC	Not mentioned
**Khansari et al., 1984** [46]	Not directly assessed, ↑ glucose uptake	Not assessed	sAD	B-Lymphocytes T-Lymphocytes	12 sAD 16 Controls
**Pansarasa et al., 2018** [106]	↓	Not assessed	sALS fALS (SOD1, TDP-43, FUS)	Lymphoblastoid cell lines	4 Controls 4 sALS 3 SOD1 2 TDP-43 2-FUS
**Smith et al., 2018** [69]	↑	↓ PDK1 and LDHB	PD	Lymphocytes and Monocytes	14 Controls 15 PD 13 RBD
**Yao et al., 2011** [66]	No change		PINK1 -/- Mice	Muscle	
**Steyn et al., 2020** [89]	↓	Not assessed		Muscle	
**Dodge et al., 2013** [96]	↑	Not assessed	Mice with SOD1	Muscle and Liver	12 Controls 11 SOD1
**Palamiuc et al., 2015** [95]	Not assessed	↑ PDK4, ↑ phospho-GS (gene expression)	Mice with SOD1	Muscle	10 Controls 13 SOD1
**Marini et al., 2020** [97]		↓ PFK	Mice with SOD1	Muscle	15 Controls 15 SOD1

This table displays the glycolysis changes seen in each of the 3 neurodegenerative diseases reviewed in this Article and cell types that the glycolysis change was seen in. The table displays enzyme changes and the number of cell lines in each article. *Abbreviations*: sAD sporadic Alzheimer’s disease, fAD familial Alzheimer’s disease, sPD sporadic Parkinson’s disease, sALS sporadic Amyotrophic Lateral sclerosis, RBC Red blood cells, PFK phosphofructokinase HK Hexokinase, PDK pyruvate dehydrogenase kinase, LDH Lactate dehydrogenase kinase. ↑ indicates and increase, ↓ indicates a decrease, and (-) indicates no change.

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
