# Peer review of "Peripheral Glycolysis in Neurodegenerative Diseases"

_ijms, 2020, doi:10.3390/ijms21238924_

Round 1

Reviewer 1 Report

This manuscript is a review of current knowledge regarding alterations in glycolysis of the peripheral cells of patients with neurodegenerative disease. The review is a good discussion of the recent literature on the topic. The manuscript is well organized and quite well written, requiring only minor revisions. 

Minor Revisions

line 75: remove capitals on Red Blood Cells

line 78: recommend changing "less" with "fewer"

line 81: add comma after (FDG-PET)

line 97: remove capital from Dementia

line 106: remove capital from Dementia

line 107: define post mortem before using acronym

line 129: replace inefficacy with inefficiency or dysfunction

line 149-150: variability in enzyme function may be mutation dependant in familial AD

line 153-155: revise run-on sentence

line 165: RBC already defined earlier in paper

line 169: add comma following "levels"

line 181-182: Since amyloid-b accumulates on brain side of BBB, more explanation required for potential RBC exposure. Mutations in mtDNA might also be another alternative.

line 186-189: revise run-on sentence

line 193: change "that" to "than"; change "one" to "1"; change "displays" to "illustrates"

line 196: add "both" after "in", and remove "seen"

line 199-202: revise run-on sentence

line 237: reference needed

line 239: remove capitals from Single Photon Emission Computed Tomography

line 246: capitalize parkinsonian

line 247: remove capital on Leptin  

line 268: sentence needs reference

line 277: add "that" after "showed"

line 336: change "eludes" to "alludes"

line 346: add space between "in" and "patients"

line 367: remove capital from Deoxyribonucleic

line 413: capitalize "complex"

line 448: add comma after realm

line 454: change "to create" to "to develop"

line 471: change "is" to "could be"

line 474: it is important to clarify that before glycolysis can be developed as a biomarker, it is important to note that changes in glycolysis must be correlated to changes linked to disease stage, progression and treatment response

Author Response

We thank the reviewer for their comments, and have made the changes they suggested as detailed below;

This manuscript is a review of current knowledge regarding alterations in glycolysis of the peripheral cells of patients with neurodegenerative disease. The review is a good discussion of the recent literature on the topic. The manuscript is well organized and quite well written, requiring only minor revisions. 

We thank the reviewer for their comment. 

Minor Revisions

line 75: remove capitals on Red Blood Cells Changed

line 78: recommend changing "less" with "fewer" Changed

line 81: add comma after (FDG-PET) Changed

line 97: remove capital from Dementia Changed

line 106: remove capital from Dementia Changed

line 107: define post mortem before using acronym Changed

line 129: replace inefficacy with inefficiency or dysfunction Changed

line 149-150: variability in enzyme function may be mutation dependant in familial AD Changed

line 153-155: revise run-on sentence Changed

line 165: RBC already defined earlier in paper Changed

line 169: add comma following "levels" Changed

line 181-182: Since amyloid-b accumulates on brain side of BBB, more explanation required for potential RBC exposure. Mutations in mtDNA might also be another alternative. MtDNA added as an explanation. AB does accumulate in the brain side, but there is also increased passage of amyloid in to the blood in AD, this has been added.

line 186-189: revise run-on sentence Changed

line 193: change "that" to "than"; change "one" to "1"; change "displays" to "illustrates"  Changed

line 196: add "both" after "in", and remove "seen" Changed

line 199-202: revise run-on sentence Changed

line 237: reference needed Added

line 239: remove capitals from Single Photon Emission Computed Tomography Changed

line 246: capitalize parkinsonian Changed

line 247: remove capital on Leptin  Changed

line 268: sentence needs reference Added

line 277: add "that" after "showed" Changed

line 336: change "eludes" to "alludes" Changed

line 346: add space between "in" and "patients" Changed

line 367: remove capital from Deoxyribonucleic Changed

line 413: capitalize "complex" Changed

line 448: add comma after realm Changed

line 454: change "to create" to "to develop" Changed

line 471: change "is" to "could be" Changed

line 474: it is important to clarify that before glycolysis can be developed as a biomarker, it is important to note that changes in glycolysis must be correlated to changes linked to disease stage, progression and treatment response Added to end of paragraph.

Reviewer 2 Report

Results and conclusions from the large cohort of studies addressing metabolic dysfunction in patients and animal models with neurodegenerative diseases (AD, PD, ALS) are often conflicting. The review article from S.M.Bell and coauthors is addressing these contradictory results in bioenergetic dysfunction PBMCs. Review is well written and structured.

Major concern:

While this review is primarily focused on PBMCs, I believe that it could benefit if changes in the glucose metabolism in peripheral cells are discussed in the context of bioenergetic status of neurons and glia. I would suggest including additional reports (PMIDs: 29055815, 29055815, 30208076), which are more focused on changes of neuronal glucose metabolism in ND.

Minor Concern:

Table 1: Signs ↓,  ↑ and (-) should be defined.

Author Response

We thank the reviewer for their comments and have made changes as detailed below;

Results and conclusions from the large cohort of studies addressing metabolic dysfunction in patients and animal models with neurodegenerative diseases (AD, PD, ALS) are often conflicting. The review article from S.M.Bell and coauthors is addressing these contradictory results in bioenergetic dysfunction PBMCs. Review is well written and structured.

We thank the reviewer for their comment

Major concern:

While this review is primarily focused on PBMCs, I believe that it could benefit if changes in the glucose metabolism in peripheral cells are discussed in the context of bioenergetic status of neurons and glia. I would suggest including additional reports (PMIDs: 29055815, 29055815, 30208076), which are more focused on changes of neuronal glucose metabolism in ND.

We thank the reviewer for their comment and have added both papers to the manuscript. The special issue of the International Journal of Molecular Sciences that this paper has been submitted to focuses mainly of peripheral markers of neurodegenerative disease, so we wanted to focus the article on this area. We have made this clearer in lines 54-55. We agree that the paper would be improved by the comparison of  peripheral and nervous system glycolysis. We have included sections were comparisons between peripheral and central glycolysis are discussed, in AD; lines 140-143, and PD; lines 316-322. We feel this has already been done in the ALS section of the paper.

Minor Concern:

Table 1: Signs ↓,  ↑ and (-) should be defined. Added

Round 2

Reviewer 2 Report

My concerns have been fully addressed.